# *Hibiscus rosa-sinensis* Forage as a Potential Feed for Animals: A Review

**DOI:** 10.3390/ani12030288

**Published:** 2022-01-25

**Authors:** Manuel Valdivié, Yordan Martínez

**Affiliations:** 1National Center for Laboratory Animal Production, Santiago de las Vegas, Rancho Boyeros, P.O. Box 6240, La Habana 10100, Cuba; mvaldivie@ica.co.cu; 2Poultry Research and Teaching Center, Agricultural Science and Production Department, Zamorano University, Valle de Yeguare, San Antonio de Oriente, Francisco Morazan, P.O. Box 93, Tegucigalpa 11101, Honduras

**Keywords:** Chinese hibiscus, green foliage, hay, chemical composition, secondary metabolites, medical properties, feeding, animal

## Abstract

**Simple Summary:**

*Hibiscus rosa-sinensis* is a medicinal plant recognized for its pharmacological chemical compounds for humans and animals. *Hibiscus rosa-sinensis* hyperproducer forage technology yields 10–20 t DM/ha/year (t of dry matter/ha/year), depending on the agronomic conditions. Chemical composition of hibiscus forage meal is similar to alfalfa, moringa, tithonia, and cassava forages due to its content of protein, minerals, fiber compound, and metabolizable energy. The fresh forage of *Hibiscus rosa-sinensis* could be offered in a ratio of 60% plus 40% of grass forage in sheep. Moreover, goats show favorable productive results when fed ad libitum with hibiscus forage. Likewise, inclusion levels of *Hibiscus rosa-sinensis* forage were recommended for rabbits, pigs, and organic chickens. Hibiscus forage could be considered as a viable alternative feed for ruminant and non-ruminant animals, although more research is needed in other animal species and productive categories.

**Abstract:**

This review summarized for the first time the nutritional benefits of *Hibiscus rosa-sinensis* in animal feed. the agronomic characteristics for the hyperproduction of *Hibiscus rosa-sinensis* were reported, as well as the chemical composition of the forage meal (leaves + petioles + stems), with emphasis on crude protein, amino acids, minerals, neutral detergent fiber, acid detergent fiber, lignin detergent acid, secondary metabolites, and metabolizable energy for ruminants and non-ruminants. Additionally, their medicinal properties such as antimicrobial, antifungal, antiparasitic, antioxidant, anti-inflammatory, antidiarrheal, and antipyretic properties were detailed. Its use as a source of protein in ruminant feeding is attractive and can be combined with grasses. In rabbits, fresh forage in high concentrations in the diets reduces the feed palatability, which causes a decrease in animal response, although it depends on the forage quality. In addition, limits for *Hibiscus rosa-sinensis* forage were recommended in diets or feeding systems for poultry, pigs, rabbits, goats, and sheep.

## 1. Introduction

*Hibiscus rosa-sinensis* is likely to have its origins in South China and tropical Asia. Today it is widely distributed throughout the Americas, Africa, Asia, and many countries in Europe, where it is cultivated as an ornamental plant for its beautiful flowers and green foliage, as well as a pharmacological plant due to its ancient and modern healing properties or as animal feed due to its contributions of protein, fiber, vitamins, and minerals [1,2]. This plant belongs to the order *malvales*, family, *malvaceae*, genus *Hibiscus*, and the species *Hibiscus rosa-sinensis*, whose common names in English are Tropical hibiscus, Chinese hibiscus, and Shoe-flower plant, while in Spanish-speaking countries they are very diverse: oceano pacífico, tulipán, cayena, hibiscus, cucarda, rosa de china, rosella, poppy, carnation, flor de avispa, sinesia, papo, pavona, and blood of Christ, among others [3,4,5,6].

In tropical and subtropical climates with average temperatures of 26 °C (13–34 °C), this plant produces leaves throughout the year, which does not occur in continental or cold climates. Additionally, it can be found at elevations from sea level to 1200 m above sea level. It can be grown in neutral or slightly basic soils (optimal pH between 5.5 and 6), with good fertility, and that are not prone to flooding, with minimum rainfall of 900 mm per year. This plant is attacked by a variety of pests (caterpillars, aphids, mealybugs, spiders, whiteflies, rust, *Alternaria alternata*, and *Botrytis cinerea* among the most frequent), but without causing irreparable damage. Forages can be cut 6 months after sowing and can survive for decades, especially with the use of fertilizers for better recycling of soil nutrients [1,3,4,7,8,9,10].

The leaves of *Hibiscus rosa-sinensis* are alternate and oval, although they can vary and even have jagged edges, dark green coloring, with a shiny appearance, abundant mucilage, and high protein content [1,11]. As animal feed, *Hibiscus rosa-sinensis* could be used in grazing as a protein bank, sown in strips with grasses or in silvopastoral systems [8,12,13]. In addition, it could be used as cut and carry forage, hay, silage, or dehydrate and is preferred to be cultivated intensively with high plant densities [8,14]. Despite the medicinal and nutritional benefits, to our knowledge this is the first review that summarizes the potential of this plant to be used in the diets of non-ruminant and ruminant animals.

## 2. Methods

### Search History

The review paper includes a total of 78 bibliographic references classified as follows: 62 articles in indexed scientific journals, 13 internationally recognized scientific books, 1 PhD thesis, 1 undergraduate thesis, and 1 report from an International Scientific Congress. It is important to note that most of the scientific information available on *Hibiscus rosa-sinensis* comes from developing countries, demonstrating the importance of its use to replace imported ingredients and reduce costs in animal production.

In the agricultural section, a total of 13 references [1,3,4,5,6,7,8,9,10,11,12,13,14] were used, which support the possibilities of the production of *Hibiscus rosa-sinensis* protein forage, as a permanent plant, with high yields of quality forage, under specific edaphoclimatic conditions, having a hyper-producing forage technology as an output. According to the references found, although the hyperproduction of forage of *Hibiscus rosa-sinensis* is described, more research is still required for the use of this technology under different edaphoclimatic conditions.

The chemical characterization and nutritional value of the *Hibiscus rosa-sinensis* forage were also described based on its age and production conditions, which demonstrated its similarity with the chemical and nutritional composition of other quality protein forages, for which 26 references were used [1,6,7,8,9,10,11,12,13,14,15,16,17,18,19,20,21,22,23,24,25,26,27,28,29,30]. Likewise, a detailed characterization of the secondary metabolites of the *Hibiscus rosa-sinensis* forage was carried out using 15 bibliographic references [8,9,10,11,12,13,14,31,32,33,34,35,36,37,38], which demonstrated the high content of various secondary metabolites within the phenolic compounds and alkaloids groups, confirming the pharmacological properties identified in this forage through 31 bibliographic references [2,8,14,32,33,37,39,40,41,42,43,44,45,46,47,48,49,50,51,52,53,54,55,56,57,58,59,60,61,62,63].

Regarding the use of *Hibiscus rosa-sinensis* forage, as animal feed, 17 references were used [1,12,64,65,66,67,68,69,70,71,72,73,74,75,76,77,78], demonstrating the levels of use of this forage in the feeding of ruminants and monogastric animals, as a source of protein, vitamins, minerals, and energy. More research and scientific information are needed on the use levels in dairy cows, fattening bulls, broilers, laying hens, ducks, geese, ostriches, quail, and guinea pigs with different forms of forage presentation (hay, silage, fresh forage, protein bank, and pellet).

## 3. Results and Discussion

### 3.1. Hyper-Producing Forage Technology from Hibiscus rosa-sinensis

Sowing is undertaken using cuttings (preferably rooted with hormones), directly in the fields or by sowing in nylon bags in nurseries (98–105 days), as after becoming rooted and developing foliage it is definitely sown in the field [9]. A density of 50,000 plants/ha has been recommended, at a distance of 1.0 m between rows and 0.20 m between plants, as well as 6 months until the establishment cut and frequent cuttings every 56 days depending on the growth speed and other factors [6], since this plant lignifies and enriches in neutral detergent fiber (NDF) and acid detergent fiber (ADF) very quickly while reducing its crude protein (CP) content [1,14,15,16]. Thus, it is advisable to cut between 30 and 45 days of regrowth in the rainy season and between 45 and 60 days of regrowth in the dry season [1,8,14,15].

On the other hand, Bolio and Sanginés [6] have reported 50 cm from the ground for manual cutting and for mechanized cuts a height between 5 and 10 cm from the ground, as is classically done with *Moringa oleifera* and *Tithonia diversifolia* [16,17]. Additionally, these authors mentioned a fertilization of 600 kg of N/ha/year, with yields between 10 and 20 t of dry matter/ha/year. A summary of this technology is shown in Table 1.

It should be noted that there are few economic studies on the production cost of *Hibiscus rosa-sinensis*, which is a decisive element for generalizing this plant in different production systems. Generally, the production cost is assumed to equal that of other protein forage plants with similar agronomic characteristics, such as *Tithonia diversifolia*, *Morus alba,* and *Trichanthera gigantea*. However, it is necessary to determine the economic feasibility of the *Hibiscus rosa-sinensis* forage hyperproduction technology system.

### 3.2. Chemical Composition

The chemical composition of *Hibiscus rosa-sinensis* forage varies depending on the aerial part of the plant used in animal feed, since the leaves with or without petioles contain the highest crude protein concentration and the lowest NDF and ADF values, when compared with the stem that is very low in CP and high in fibrous component, having an intermediate value for the aerial part, which is the part most used in animal feeding [1].

Table 2 shows the chemical composition of the total forage, made up of all the leaves + petioles + stems. It is clearly observed that as the plant ages it increases dry matter (DM), NDF, and ADF, and decreases CP in its aerial parts.

The effect of the hibiscus forage cutting age, on the forage chemical composition (leaves + petioles + stems) is shown in more detail in Table 3, which considers earlier cutting ages (30–45 days). Additionally, older forage increases the DM/ha yield, but at the expense of low protein and high ADF contents, which reduce the total digestible nutrient (NDT) contributions for ruminants and metabolizable energy (ME) for poultry and pigs. It is advisable to use forages cut between 30 and 45 days in animal feed due to their better nutrient content and commercial quality.

The average, minimum, and maximum of the chemical composition of forage meal are shown in Table 4. The values are influenced by age, genotype, soil fertility, cut, environmental characteristics, and other sources of variation that influence the chemical composition of protein forages [18,19,20,21,22,23]. The average value is similar to that of a regrowth of around 45 days (Table 3).

*Hibiscus rosa-sinensis* forage is comparable to the chemical composition of alfalfa forage reported by NRC [15], INRA [16], FEDNA [18], and Heuze et al. [19] and the forages of *Moringa oleifera* [20,21], *Tithonia diversifolia* [22], *Morus alba* [23], *Trichantera gigantea* [24,25], and cassava meal [26]. This forage (Hibiscus) also has a higher concentration of crude protein than most of the cereals commonly used in animal nutrition, such as corn, sorghum, millet, oats, barley, rye, triticale, and common wheat, but with lower concentrations of ME than those of all of the mentioned cereals [15,16,17,18]. These results demonstrate the great potential of *Hibiscus rosa-sinensis* as a protein forage to be used in the feeding of ruminant and monogastric animals.

In this sense, the ME and digestible energy (DE) values of hibiscus forage meal for ruminants, rabbits, and guinea pigs are shown in Table 4, which are appropriate for these animals and indicate the possibility of using them in high proportions in their diets. However, the low contribution of ME for poultry and pigs demonstrates its low potential to replace cornmeal and other energy sources in the diets of these non-herbivorous monogastric animals [15,16,17,18,19,20].

Table 5 presents the amino acid profile of *Hibiscus rosa-sinensis* forage, showing all essential amino acids. *Hibiscus rosa-sinensis* forage has higher contents of lysine and threonine than alfalfa forage [16,19], therefore, the partial or total substitution by the proposed forage (*Hibiscus rosa-sinensis*) could increase the contributions of these essential amino acids in the diets.

In animal nutrition, lysine is recognized as the reference amino acid and has an important role in the absorption of calcium and in the production of antibodies, hormones, and enzymes. Additionally, lysine-rich diets can stimulate egg production and breast yield in poultry [27]. In addition, threonine, which is an essential amino acid, interferes with intestinal permeability, fat metabolism, and hepatic glucose synthesis, which favors animal performance and the end product of zootechnical productions (meat and egg) [28,29].

However, in this forage low concentrations of arginine, tryptophan, and alanine were detected and the levels of methionine plus cystine were similar to those of alfalfa forage [16,17,18,19], therefore, these amino acids should be included synthetically in the diets of monogastric animals fed high concentrations of hibiscus forage. These results confirm that *Hibiscus rosa-sinensis* forage has an acceptable content of the three main essential and limiting amino acids (lysine, methionine, and threonine) in monogastric animals, which could also partially replace soybean meal and other protein sources commonly used in animal feed, such as alfalfa meal for ruminant nutrition.

### 3.3. Secondary Metabolites

The main secondary metabolites of *Hibiscus rosa-sinensis* according to Jadha et al. [31] are flavonoids (cyaniding-3-sophoroside-5-glucoside, kaempferol-3xylosylglucoside, cyaniding-3,5-diglucoside, quercetin-3-sophorotrioside, quercetin-3,7-diglucoside, quercetin-3-diglucoside) with recognized antioxidant and pharmacological properties [31,32].

In other phytochemical studies carried out by Hernández et al. [8], Sobhy et al. [33], and Riascos-Vallejos [14] on forages and leaves of *Hibiscus rosa-sinensis*, these authors reported that this plant has abundant flavonoids in the form of catechins with recognized antioxidant properties [34]; pigmenting quinones with cytotoxic properties that are used against some types of cancer [35,36]; triterpenes, with high antioxidant capacity, that are capable of inhibiting diabetes, embryopathies, and neuropathies, although they can also promote the synthesis of steroids when they appear in the form of squalene [37]. In the phytochemical studies carried out using the leaves of *Hibiscus rosa-sinensis*, mucilages have also been found, which are colloidal agents that are found within the water-soluble dietary fiber, can produce gels, and in adequate concentrations they can benefit the growth of the intestinal microflora and hence the animal response [38,39]. Likewise, a moderate content of amides in this plant (*Hibiscus rosa-sinensis*) that provide hormonal balance through vitamins, constitutes the nitrogenous bases of DNA and RNA, and the amino groups of amino acids that through peptide bonds form proteins and enzymes are extremely vital [40]. The contents of phenols (2.61%), condensed tannins (0.3 to 1.9%), alkaloids (0.87%), and saponins (0.09%) reported by Hernández et al. [8] and Riascos-Vallejos [14] for *Hibiscus rosa-sinensis*, may be associated with the low palatability detected in fattening rabbits and guinea pigs with the use of this forage.

### 3.4. Pharmacological Properties

*Hibiscus rosa-sinensis* is a prominent medicinal plant, with varied effective and safe pharmacological activity [2,32,41]. Leaves showed antibacterial properties against *Staphylococcus aureus*, *Staphylococcus aerogenes*, *Bacillus subtilis*, *E. coli,* and Shigella dysenteriae [42,43,44,45,46,47] and antifungal properties against *Aspergillus niger*, *Aspergillus flavus*, *Candida albicans*, *Candida parapsilosis,* and *Candida glabreta* [43,48,49]. Leaves of *Hibiscus rosa-sinensis*, demonstrated also antiparasitic properties in vitro and in vivo against *Hymenolepis diminuta* [50]. Despite the anticestodal and antifungal properties in the different extracts of the leaves of *Hibiscus rosa-sinenesis*, the inclusion level of the forage of this plant capable of demonstrating these pharmacological properties is still unknown.

Leaves of *Hibiscus rosa-sinensis* have abundant phenolic compounds (48.4 mg equivalent to catechin/g of DM) and flavonoids (equivalent to 24.26 mg of quercetin/g of DM), promoting excellent antioxidant activity [33,51,52,53]. The aqueous extracts obtained from the leaves of *Hibiscus rosa-sinensis* administered between days 1 and 6 of gestation inhibit implantation in the endometrium due to the increase in superoxide anionic radicals and an excessive decrease in the enzyme superperoxidase dismutase, which causes a reduction in estrogenic activity [51]. Additionally, this medicinal plant has been used to cause abortions and to stimulate the expulsion of the placenta after animal births [32,54].

In future research, it is important to elucidate at what reproductive age the forage intake of *Hibiscus rosa-sinensis* provokes the hormonal changes and reproductive damage in the previously mentioned reproductive females [32,55].

The ethanolic extract of *Hibiscus rosa-sinensis* leaves has a potent anti-inflammatory effect in edematous syndrome caused by carrageenan injection, which was associated with a reduction in polymorphonuclear leukocyte infiltration [53]. In a study with male Wistar rats, where colitis was induced by injecting 2 mL of 4% acetic acid intrarectally, the hydroalcoholic extract of the leaves of *Hibiscus rosa-sinensis* significantly reduced ulcers, oxidative stress, and other damage in the colon, which corroborates the anti-inflammatory properties of this medicinal plant [56].

In another study, the leaves of *Hibiscus rosa-sinensis* were evaluated to control cough in guinea pigs. They were used in a chamber where the cough was induced, and the extract of this medicinal plant (Hibiscus) significantly decreased the frequency of cough in the animals [57]. Furthermore, Ali et al. [58] and Mondal et al. [59] found that the aqueous and ethanolic extracts of *Hibiscus rosa-sinensis* leaves have a large healing effect in laboratory mice, respectively.

Other pharmacological properties of *Hibiscus rosa-sinensis* leaves include counteracting dysentery and diarrhea, arthritis, boils, and phlegmons [2,32,55,60]. They are also antipyretics [61], cardio-protectors [62], and antidiabetic [37,63].

#### Toxicity of Ethanolic Extracts

The ethanolic extracts of *Hibiscus rosa-sinensis* leaves show toxicity, when the animals (mice) receive a dose of 800 mg/kg of body weight for 14 days, evidenced by an increase in alanine transferase, aspartate aminotransferase, bilirubin, urea, and creatinine. Histological sections in the liver show dilated sinuosities, nuclear apoptosis, and inflammation in the capillaries of the liver, together with disorganization of the tubules and glomeruli of the kidney with thickening of the interstitial spaces of that organ [60]. Previous authors indicated that the mean lethal dose (LD50) was higher than 2000 mg/kg, thus, its use at low concentrations, such as those used in the different pharmacological tests mentioned, make it safe for animal welfare.

### 3.5. Hibiscus rosa-sinensis as Animal Feed

There is ample empirical and scientific information on the use of *Hibiscus rosa-sinensis* as an ornamental and pharmacological plant; however, the data on its use in animal nutrition are generally empirical and poorly validated by science. This review shows the results supported by scientific research, which could contribute to promoting the research, production, and use of hibiscus forage in animal nutrition.

#### 3.5.1. Sheep Feeding

Currently, in several regions where sheep are raised, the tree species used for forage are frequently used in diets [64]. *Hibiscus rosa-sinensis* is a shrubby plant, capable of replacing the commercial concentrate in diets of growing hair sheep with an inclusion level of 1% of body weight or 167 g of dry matter/animal/day [65], the result of the combination of nutrients, digestibility of dry matter (DM), and preference of sheep towards this forage. These authors found that the supplementation of grazing lambs with hibiscus meal did not modify the ruminal pH values, the ammonia nitrogen concentration, nor the molar proportion of volatile fatty acids.

Likewise, Obrador-Olán et al. [66] found that supplementation with *Hibiscus rosa-sinensis* forage in fattening sheep after grazing (star grass) increased daily gain by 46.7% compared to the group that only consumed star grass during grazing (73 vs. 107 g/sheep/day), which demonstrated the potential of this protein forage (*Hibiscus rosa-sinensis*) for sheep in grasslands. Ruiz-Sesma et al. [4] found that a ratio of 60% *Hibiscus rosa-sinensis* forage and 40% star grass forage, significantly promoted the body weight gain of 125 g/young sheep/day. This increase in gain turned out to be 2.7 times greater than the diet based exclusively on star grass. Additionally, 60% of hibiscus forage stimulated the intake of dry matter and increased the digestibility of DM, CP, and ADF. Mata-Espinosa et al. [67] demonstrated that the 85% star grass + 15% hibiscus system promoted greater growth speed than the 84% star grass + 16% *Morus alba* and 89% star grass + 11% *Gliricidia sepium* systems.

The feeding of Pelibuey sheep stallions, with grass plus a concentrate of 16% CP, was replaced by a feeding with grass plus the daily supplementation of 1.6 kg of *Hibiscus rosa-sinensis* forage without affecting the volume or sperm concentration of the ejaculate, together with higher indices of individual motility (average speed, rectilinear speed, and curvilinear speed), which showed that *Hibiscus rosa-sinensis* is an alternative product for use in the feeding of Pelibuey sheep stallions in the tropics, without affecting semen quality [67].

Likewise, Aguilar-Urquizo [64] compared a control group (balanced feed grains plus star grass), mulberry group (mulberry foliage plus balanced feed), and hibiscus group (hibiscus foliage plus balanced feed) in Pelibuey male sheep. They found that the use of hibiscus foliage decreased feed intake, although the intake of phytoestrogens and tannins was similar compared to the mulberry group, and both groups did not affect the onset of puberty. These authors concluded that both experimental diets can be used in breeding sheep without affecting reproductive indicators.

#### 3.5.2. Goat Feeding

It is known that goats, unlike sheep, have a feed preference for certain forages [68]. In this sense, in a study carried out by Nhan [69] in Vietnam with growing goats, they found that ad libitum supply of fresh forage of low quality *Hibiscus rosa-sinensis* (leaves plus thin stems), with 90 days of regrowth and low protein content (13.82%), provoked a body weight gain of 77 g/goat/day and a fresh forage intake of 2.49 kg/goat/day, when this plant (*Hibiscus rosa-sinensis*) was the only feed used. This result demonstrated the nutritional potential of *Hibiscus rosa-sinensis* forage as feed for goats, although more studies are needed to determine the feeding efficiency of this forage (Hibiscus) in the different productive categories in this small ruminant (goat). Apparently, due to the feeding system of goats, the use of hibiscus forages with fewer days of regrowth (30–60 days) could favor the animal response due to the higher concentration of crude protein (16.6–19.6%; Table 2 and Table 3) and lower content of fibrous compounds (Table 2 and Table 3), as well as other antinutritional factors associated with fiber such as lignin [1,8,10].

#### 3.5.3. Rabbit Feeding

The use of protein forages in rabbit feeding is a common practice that contributes to cecotrophy and production of cecotropes and therefore in the productive response of this monogastric herbivorous animal [1]. Martínez et al. [12] evaluated different forms of supply of *Hibiscus rosa-sinensis* forage to fattening rabbits and appreciated that when feed is offered in a restricted way, rabbits do not consume enough fresh hibiscus forage, as occurs with other forages rich in proteins. However, when feed was offered ad libitum, the rabbits consumed 25 g/DM of this forage and had a body weight gain similar to the control group that consumed feed without forage and to another treatment with ad libitum feed plus *Brosimum alicastrum* forage.

Furthermore, Lara et al. [70] compared two mini-blocks with 30% mulberry meal and 27% *Hibiscus rosa-sinensis* meal, they found that the second block (*Hibiscus rosa-sinensis*) significantly reduced feed intake (153 vs. 122 g/day) and the daily gain in body weight (19.2 vs. 14.7 g/day), although unchanged for protein digestibility.

Likewise, La O et al. [71] proposed an alternative feeding system for rabbits, where the sugarcane stem was used ad libitum as a basic source of energy and the protein was supplied with four protein forages, such as *Hibiscus rosa-sinensis, Teramnus labialis, Phyla nodiflora,* and *Ipomea potato* plus 25 g/rabbit/day of sunflower seeds. The authors found that the viability was excellent in this experiment (100%). Additionally, the animals fed *Teramnus labialis* had the best body weight gains (23 g/rabbit/day) while the group that consumed *Hibiscus rosa-sinensis* showed the worst body weight gains (16 g/rabbit/day), which were 30% lower. This forage (Hibiscus) increased the feed conversion due to the low palatability and lower consumption of DM. Rabbits consumed 18 and 38% less CP and DE than the group with *Teramnus labialis*, respectively. It is important to highlight that *Hibiscus rosa-sinensis* forage is not very palatable, perhaps due to the high concentration of secondary metabolites, which reduces nutrient intake, body weight gain, and decreases feed efficiency and profitability of fattening rabbits [1,12,72,73].

In breeding rabbits, authors such as Canul-Ka et al. [72] have shown that it is feasible to replace 40% of the intake of the conventional diet in lactating rabbits, when they have free access to *Hibiscus rosa-sinensis* leaves without affecting their productive response, which is measured through milk production and the growth of the rabbits. Additionally, the calving interval was not affected which represented a saving of 39% in production costs. When the restriction of the basal diet is higher than 50%, the lactating rabbits consume more than 200 g/d/fresh hibiscus foliage. In this study, the anti-implantation and abortifacient effects attributed to this plant by Jadhav et al. [31], Nivsarkar et al. [53], and Chopra et al. [54] were not evidenced.

In growing male rabbits (future breeders), from 35 to 210 days of age, the ad libitum supply of fresh forage of *Hibiscus rosa-sinensis* allowed to substitute up to 60% of the commercial balanced feed, without affecting the growth rate or producing macroscopic alterations in the main reproductive organs of male rabbits, which had a positive economic impact [73].

#### 3.5.4. Organic Chicken Feeding

At present, many companies are interested in producing organic food products in order to benefit consumer health and generate added value to the product, however, the most suitable feeding systems for the animals produced within the system are still being studied [77]. According to the USDA [78] guidelines for organic certification of poultry, broilers throughout their productive life must be fed cereals and forages that have been produced under organic conditions. In this sense, *Hibiscus rosa-sinensis* (under organic production) leaf meal from 45 days of regrowth was efficiently used in the diets for organic chickens from birth to 12 weeks old, at levels of 5 and 10%, which improved body weight at slaughter, feed conversion ratio, carcass pigmentation, and economic profitability, without affecting the edible portion yields [74]. These results indicate the possibility of using this forage meal at low levels in poultry diets. However, more research is needed to know the yield of *Hibiscus rosa-sinensis* under organic farming, as well as studies in fast-growing broilers and laying hens, to demonstrate the efficacy of this plant (*Hibiscus rosa-sinensis*) in different production systems.

#### 3.5.5. Swine Feeding

Pigs, due to their omnivorous behavior, accept within their ration a wide variety of feeds, including protein forages, which are alternative feeds used to reduce the production cost associated with feeding [75]. Thus, Ly [76] reported that the inclusion of 10 and 20% *Hibiscus rosa-sinensis* forage meal in growing pigs did not modify the ileal digestibility of DM (76%, 74%, and 73%) and CP (64%, 61%, and 62%). Ileal digestibility of OM (82%, 78%, and 76%) and energy (79, 73, and 71%) had slight decreases and ileal digestibility of ashes increased (43%, 43%, and 47%). This author also recommended the inclusion of between 10 and 20% of *Hibiscus rosa-sinensis* meal in the diets of finisher pigs and lactating sows, although it is necessary to adjust the contribution of digestible energy in the diets because these categories require greater concentrations (digestible energy). Furthermore, according to the results of nutrient digestibility [69] and the levels of use in growing rabbits [12], breeding rabbits [70], and organic chickens [74], it appears that dietary inclusion of up to 10% of this forage does not decrease voluntary consumption. However, more research is needed with different forms of presentation of *Hibiscus rosa-sinensis* in different categories of pigs.

#### 3.5.6. Inclusion Levels

Table 6 summarizes the recommended inclusion levels of *Hibiscus rosa-sinensis* forage in animal feed, whether in the form of fresh forages, meal, or hay, prepared from the information utilized in this review.

To our knowledge, this is the first review that summarizes the nutritional benefits of *Hibiscus rosa-sinensis* forage, which could contribute to its more widespread use in animal nutrition. However, it is necessary to generate more scientific and practical information on the use of hibiscus forage at different cutting ages in animal feed (cattle, goats, poultry, and pigs), deepening its palatability, chemical composition, and secondary metabolites. It is also necessary to investigate the anti-implantation and abortifacient effects attributed to this plant, especially in animals fed ad libitum and in fresh form, as well as to carry out experiments with high concentrations of fresh forage, silage, hay, or granules of *Hibiscus rosa-sinensis* for a long period of time, to determine whether the concentration and time can cause any type of damage.

## 4. Conclusions

The hyperproductive forage technology of *Hibiscus rosa-sinensis* (50,000 plant/ha) can obtain yields between 10 and 20 t DM/ha year. The chemical composition of *Hibiscus rosa-sinensis* forage meal in CP, essential amino acids, NDF, ADF, ME, calcium, and phosphorus is similar to the chemical composition of alfalfa forage and other alternative protein forages used in animal nutrition. *Hibiscus rosa-sinensis* is a promising medicinal plant, with a wide variety of beneficial secondary metabolites which at low concentrations show effective and safe pharmacological activity.

Scientific reports on the use of *Hibiscus rosa-sinensis* as feed show that fresh hibiscus forage can be offered to growing sheep at a rate of 1% of their body weight plus grass forage or in a proportion of 60% of hibiscus forage plus 40% of grass forage. Likewise, goats can consume *Hibiscus rosa-sinensis* forage ad libitum as the only feed, which promotes a moderate daily gain. The use of this forage ad libitum is recommended in rabbit diets, without reducing the productive and reproductive indicators. Additionally, 10% of hibiscus forage meal in pig diets did not affect nutrient digestibility and the inclusion of 5–10% of this meal improved the productive response of broilers raised under an organic system.

## Figures and Tables

**Table 1 animals-12-00288-t001:** Summary of *Hibiscus rosa-sinensis* hyperproducer forage technology. DM: dry matter.

Items	Details
Plantation density (50,000 plants/ha)	1.0 m between rows and 0.20 m between plants
Staggered planting density for strip planting with grasses	1.0 m between double rows and 0.20 m between plants
Establishment cut, months	6
Subsequent cuts in rainy season, days	30–45
Subsequent cuts in dry season, days	45–60
Machined cutting height, cm	10
Manual cutting height, cm	50
N, kg/ha/year	Minimum 600 (organic is better)
P, kg/ha/year	Minimum 150
K, kg/ha/year	Minimum 150
Irrigation	Required for all
Maximal forage yield, t DM/ha/year	20
Minimal forage yield, t DM/ha/year	10

Sources: La O [1], Ruiz-Sesma et al. [4], Bolio and Sanginés [6], Cuellar and Arrieta [7], and Hernández et al. [8].

**Table 2 animals-12-00288-t002:** Chemical composition of the aerial parts of *Hibiscus rosa-sinensis* cut at two ages.

Aerial Parts of the Plant	Age	
(100% DM)	60 Days	120 Days
DM in leaves, %	19	28
DM in stems, %	23	42
DM in leaves + stems, %	20	28
CP in leaves, %	20.7	14
CP in stems, %	7.9	5.9
CP in leaves + stems, %	16.6	10.8
NDF in sheets, %	42	56
NDF in stems, %	52	69
NDF in leaves + stems, %	45	61
ADF in sheets, %	29	35
ADF in stems, %	31	47
ADF in leaves + stems, %	30	39

Source: La O [1]. DM: dry matter; CP: crude protein; NDF: neutral detergent fiber; ADF: acid detergent fiber.

**Table 3 animals-12-00288-t003:** Effect of regrowth cutting age on DM yield and chemical composition (100% DM) of forage.

	Ages (Days)						
Leaves + Petioles + Stems	30	45	60	75	90	105	120
Yield, t of DM/ha/year	9.4	12.1	15.3	18.2	20.5	-	-
DM, %	18.9	19.6	20.0	22.0	24.0	28.0	28.0
CP, %	19.6	17.3	16.6	15.0	14.2	12.6	10.8
NDF, %	33	38	45	50	53	56	61
ADF, %	23	27	30	32	34	36	39
LAD, %	7.5	9.0	9.4	11.1	13.5	14.0	14.2

Sources: La O [1], Hernández et al. [8], and Cruz et al. [9] DM: dry matter; CP: crude protein; NDF: neutral detergent fiber; ADF: acid detergent fiber; LAD: lignin acid detergent.

**Table 4 animals-12-00288-t004:** Chemical composition of forage meal (leaves + petioles + stems) on a dry basis (100% DM).

Items	Average	Minimum	Maximum
Crude protein, %	17.02	11.32	19.72
Ether extract, %	4.19	3.5	5.2
Ash, %	13	11	17
Nitrogen-free extract, %	11.1	10	11.9
Crude fiber, %	17.7	15	22
NDF, %	38.44	31	53.47
ADF, %	27.04	17.3	34.3
LAD, %	9.2	5.7	13
Organic material, %	87	83	89
Calcium, %	1.05	0.55	3.35
Total phosphorus, %	0.3	0.2	0.52
Magnesium, %	0.21	ND	ND
Sulfur, %	0.08	ND	ND
Manganese, %	3.1	ND	ND
Iron, %	0.24	ND	ND
Copper, %	4.1	ND	ND
Zinc, mg/kg	27.82	ND	ND
In vitro degradability of DM, %	80.66	ND	ND
In vitro degradability of OM, %	76.77	ND	ND
In vitro digestibility of DM, %	71.7	51.3	79.2
Gross energy, kcal/kg DM	4068	3864	4229
DE rabbits, kcal/kg	2380	1987	2875
ME ruminant, kcal/kg	2150	ND	ND
ME poultry, kcal/kg	1750	ND	ND
ME pigs, kcal/kg	1900	ND	ND
ME guinea pig, kcal/kg	2400	ND	ND

Sources: La O [1], Bolio and Sanginés [6], Hernández et al. [8], Cruz et al. [9], Flores et al. [10], Leyva et al. [11], Martínez et al. [12], Meza et al. [13] and Riascos-Vallejos et al. [14]. DM: dry matter; NDF: neutral detergent fiber; ADF: acid detergent fiber; LAD: lignin acid detergent. OM: organic material; DE: digestible energy; ME: metabolizable energy; ND: not determined.

**Table 5 animals-12-00288-t005:** Amino acid content of the forage meal (leaves + petioles + stems) of *Hibiscus rosa-sinensis* (100% DM).

Amino Acids, %	Forage (%)
Lysine	1.1
Methionine	0.3
Cystine	0.18
Threonine	0.86
Tryptophan	0.22
Arginine	0.56
Valine	0.52
Isoleucine	0.56
Leucine	1.13
Phenylalanine	3.55
Tyrosine	1.73
Glycine	0.10
Serine	0.60
Proline	0.91
Histidine	0.3
Alanine	0.75
Aspartic acid	1.14
Glutamic acid	1.12
Amino acid total, %	15.62
Crude protein, %	17.02

Source: Gutiérrez et al. [30]. DM: dry matter.

**Table 6 animals-12-00288-t006:** Forms and levels of use of hibiscus forage in animal feed reported in the literature.

Animal Category	Forms of Use in Animal Feed	Sources
Sheep	60% of hibiscus forage plus 40% of star grass forage	Ruiz-Sesma et al. [4]
Sheep	1% of body weight as fresh hibiscus forage plus other forages	Mata-Espinosa et al. [65]
Goats	Hibiscus forage ad libitum	Nhan [68]
Growing rabbits	25 g of *Hibiscus rosa-sinensis* as feed	Martínez et al. [12]
Rabbit breeding	Hibiscus forage ad libitum plus 60% of a conventional diet	Canul-Ku et al. [71]
Male rabbits (35–210 days old)	Hibiscus forage ad libitum plus 40% of a conventional diet	Ramos-Canché et al. [72]
Organic chickens	5–10% of hibiscus forage in diets	Saltos [74]
Growing pigs	10% of hibiscus forage in diets according to digestibility information	Ly [76]
Lactating sows	10% of hibiscus forage in diets	Ly [76]

## Data Availability

Not applicable.

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
