# Peer review of "Hibiscus rosa-sinensis Forage as a Potential Feed for Animals: A Review"

_animals, 2022, doi:10.3390/ani12030288_

Round 1

Reviewer 1 Report

  1. According to the manuscript, Hibiscus rosa-sinensis is a great potential feed for animals, however, the advantage composition of Hibiscus rosa-sinensis than other feeds have not been discussed and the comparison of feeding efficiency with other feeds also should be reviewed.
  2. Meanwhile, this manuscript just provided the yield, the capital input and cost should not be ignored. Furthermore, the growth time of Hibiscus rosa-sinensis is not short, how to expand industrial scale?
  3. On the other hand, Hibiscus rosa-sinensis is a prominent medicinal plant, what about the medicinal composition? Whether long-term consumption is harmful?

Author Response

Dear reviewer,

Thank you very much for your comments on our manuscripts.

Comments and Suggestions for Authors

Reviewer: According to the manuscript, Hibiscus rosa-sinensis is a great potential feed for animals, however, the advantage composition of Hibiscus rosa-sinensis than other feeds have not been discussed and the comparison of feeding efficiency with other feeds also should be reviewed.

Authors: Done.  “Chemical composition of Hibiscus rosa-sinensis forage meal with emphasis in crude protein, amino acid profile (Table 4 and 5), NDF, ADF, ME, calcium and phosphorus are shown in tables 4 and 5. This forage is comparable to the chemical composition of alfalfa forage reported by NRC [16], INRA [17], FEDNA [19] and Heuze et al. [20] and to forages from Moringa oleifera [21,22], Tithonia diversifolia [23], Morus alba [24], Trichantera gigantea [25,26] and cassava meal [27], as well as a higher content of crude protein a of most cereals commomly used in animal nutrition as well corn, sorghum, millet, oats, barley, rye, triticale and common wheat, but with concentrations of ME lower than those of all the mentioned cereals [16-19]. Thus, this plant shows its potential as a protein forage to be used in ruminants and herbivores as a good source of proteins, minerals, and ME”.

Likewise, the ME and DE values of hibiscus forage meal for ruminants, rabbits and guinea pigs is shown in table 4, which are appropriate for these animals and indicate the possibility of using them in high proportions in their diets, however, the low contribution of ME for poultry and pigs demonstrates its low potential to replace maize and other sources of EM in the diets of these non-herbivorous monogastric animals [16-20].

Reviewer: Meanwhile, this manuscript just provided the yield, the capital input and cost should not be ignored. Furthermore, the growth time of Hibiscus rosa-sinensis is not short, how to expand industrial scale?

Authors: It is an interesting question. Unfortunately, there is no precise information available on the farming costs of this plant. We incorporate a paragraph “It should be noted that there are few economic studies on the production cost of Hibicus rosa-sinensis, which is a decisive element to generalize this plants in different production systems. Generally, the cost of production is assumed to be that of other similar protein forage plants such as Tithonia diversifolia, Morus alba and Trichantera gigantea. However, it is necessary to determine the economic viability of the hyperproducing forage technology system in Hibicus rosa-sinensis.”

Reviewer: On the other hand, Hibiscus rosa-sinensis is a prominent medicinal plant, what about the medicinal composition?

Authors: As you suggest, this plant is recognized as a medicinal plant, so from line 156 to line 222 we list the main secondary metabolites and their pharmacological properties, we also emphasize the toxicity of the ethanolic extract. We think these lines summarize the medicinal properties to answer your question.

Reviewer: Whether long-term consumption is harmful?

Authors: Although the studies are not conclusive, we did indicate in the manuscript that it has been shown to be safe in sheep in tests that lasted 90 days [63, 65], in organic chickens 84 days [69] and in rabbits 175 days [69]. We incorporate a paragraph “This is the first review that summarizes the nutritional benefits of Hibiscus rosa-sinensis, which is important for its more widespread use in animal nutrition. However, it is necessary to generate more scientific and practical information on the use of Hibiscus forage at different cutting ages in animal feed (bovines, goats, poultry, and pigs), deepening its palatability, chemical composition, and secondary metabolites. It is also necessary to investigate the antimplantation and abortifacient effect attributed to this plant, especially in animals fed with ad libitum and in fresh form, as well as to carry out experiments with high concentrations of fresh forage, silage, hay, or granules of Hibiscus rosa-sinensis for a long time, to know if the concentration and time can cause any type of damage”.

Reviewer 2 Report

The manuscript „Hibiscus rosa-sinensis forage as a potential feed for animals” describes agronomic characteristics for the hyperproduction of Hibiscus rosa-sinensis, as well as the chemical composition of the forage meal. The authors pointed out the chemical composition, secondary metabolites,  pharmacological properties antifungal and toxicity of ethanolic extracts of Hibiscus rosa-sinensis which gives readers valuable information about this plant. Further, in the article Valdivié and Martínez presents the feeding potential of Hibiscus rosa-sinensis.
In my opinion, the article is well written and provides very useful knowledge. However, I have one issue for the authors: for all names of plants should be used italics.

Author Response

Dear reviewer,

Thank you very much for your comments on our manuscripts.

 Comments and Suggestions for Authors

The manuscript, Hibiscus rosa-sinensis forage as a potential feed for animals” describes agronomic characteristics for the hyperproduction of Hibiscus rosa-sinensis, as well as the chemical composition of the forage meal. The authors pointed out the chemical composition, secondary metabolites, pharmacological properties antifungal and toxicity of ethanolic extracts of Hibiscus rosa-sinensis which gives readers valuable information about this plant. Further, in the article Valdivié and Martínez presents the feeding potential of Hibiscus rosa-sinensis
In my opinion, the article is well written and provides very useful knowledge. However, I have one issue for the authors:

Reviewer: for all names of plants should be used italics.

Authors: Done

Reviewer 3 Report

Please see the attached file for my comments

Author Response

Dear reviewer,

Thank you very much for your comments on our manuscripts. Really, an excellent review work is denoted.

Reviewer

Reviewer: The authors reviewed Hibiscus rosa-sinensis forage as a potential feed for animals based on functional properties of Hibiscus. Still, the content that focuses on the nutritional properties of Hibiscus is very limited given the main topic is a potential feed for animals. When evaluating a feed ingredient for animals, the authors should present more information regarding nutrient variability, nutrient digestibility, relative value to corn, soybean meal, or other common feed ingredients fed to animals, availability in a market, anti-nutritional factors, handling properties, and inclusion rate. Because Hibiscus rosa-sinensis forage is not a common feed ingredient, it will be necessary to consider and discuss these factors when evaluating novel feedstuff.

The authors summarized the literature's information but failed to discuss their findings or area that needs to further research.

 Finally, the information on the inclusion rate of Hibiscus rosa- sinensis in animal feed solely depends on minimal studies and should not be concluded as safe inclusion levels or recommendation for inclusion.

Overall, this review manuscript is not well organized in the content flow and written without a comprehensive scientific view.

By the way, some sentences are not grammatically correct and require re-read sentences several times to infer the meaning.

 The authors should consider an English editing service for this review paper if this article is going to publish in a journal.

Authors: Done. We greatly improved the wording and gave it a more scientific approach by comparing it to similar forages and other conventional feeds that are commonly used in animal nutrition. Writings are marked within the text in yellow.

After adjusting the writing of the manuscripts according to the recommendations of the reviewers, we consider that this new version of the manuscript meets your criteria for organization, fluency, and comprehension. We consider the manuscript valid for the international scientific community because it is the first time that the nutritional benefits of hibicus forage in animal feed have been summarized. The characterization of secondary metabolites and their pharmacological properties is also described in depth, as well as the need for further studies.

Reviewer: Ln 16-17: “The maximum and safe inclusion of Hibiscus forage meal in balanced diets” is not quite accurate because the evaluation of safety of inclusion of hibiscus forage meal did not state/report in the review paper yet.

Authors: Done. The wording was improved

Reviewer: Ln 39: pharmacological plant should be changed to pharmacological properties.

Authors: Done.

Reviewer: Ln 41: the order malvales, family, malvaceae, genus hibiscus and the species Hibiscus rosa-sinensis, the hibiscus should be italic form

Authors: Done.

Reviewer: Ln 41-56: Not sure if they are related to the main context of this review paper. This information relates to more the science and technology of producing Hibiscus in agriculture, which could be the focus for Agronomy, but is not directly related to feed ingredients and animal nutrition.

Ln 67-81: Not sure if it is necessary to write two paragraphs to describe the Hyper-producing forage technology for Hibiscus rosa-sinensis, especially the authors have already provided the table to summarize the technology. Additionally, the information in the table is much more interesting and relevant to agronomists instead of animal scientists who are the main audiences for using Hibiscus fed to animals.

Authors: It is an interesting question. Certainly, the misuse of Hibicus forage or other similar ones in animal feed is due to the lack of knowledge of the chemical composition, levels of use and agronomic conditions. Generally, these three pillars must be interrelated, especially when describing a new feed, these are mentioned in the manuscript. We consider that indicating the agronomic conditions is one of the aspects that could define its most widespread use in the world; being a novelty of this manuscript, which summarizes the hyperproductive forage system adapted for the cultivation of hibicus.

A similar article, which includes agronomic conditions, was published in a high-impact journal.

Valdivié, M., Martínez, Y., Mesa-Fleitas, O., Botello-León, A., Hurtado, C. B., & Velázquez-Martí, B. (2020). Review of Moringa oleifera as forage meal (leaves plus stems) intended for the feeding of non-ruminant animals. Animal Feed Science and Technology, 260, 114338.

Reviewer: Ln 63-65: the topic “Hibiscus rosa-sinensis forage as a potential feed for animals” and how Hibiscus forage meal is relevant to current feed ingredients used in monogastric animals and ruminant animals should be introduced in the introduction. Then, it will be much more appropriate to reveal this review is the first review to summarize “the potential of this plant to be used in the diets of non-ruminant and ruminant animals.

Authors:  Done. In the manuscript, it was specified that this was the first study to summarize the nutritional benefits of Hibiscus rosa-sinensis forage in animal nutrition, and further studies are necessary.

Reviewer: Ln 88: NDF and ADF should be defined upon first use. Ln 100: NTD should be defined.

Ln 95, 109, and 117: table 3, 4 and 5: crude protein or CP should be

Ln 109: Abbreviation used in table 3 should be defined Ln 117: a footnote is needed for ND

Authors: Done

Reviewer: Ln 120: remove “too near” and consider using “comparable to”

Authors: Done

Reviewer: Ln 128: Thyroxine is a hormone not an essential AA. Please revise it.

Authors: Done

Reviewer: Ln 136-138: please provide references

Authors: Done

Reviewer: Ln 146-149: Please revise the sentence. “Mucilages with biological activity” is too general and Mucilages should not be considered as secondary metabolites

Authors: Done

Reviewer: Ln 156-159: For scientific writing. the genus name is capitalized and the species is lower case. The names should be italicized.

Authors: Done

Reviewer: Ln 159-161: Please check the reference and revise the sentence, the anticestodal effects of Hibiscus rosa-sinensis L. (Malvaceae) leaf is from methanol extraction, so feeding Hibiscus forage meal 5 to 10% to livestock may not exhibit the anticestodal effect.

Authors: We read again the article entitled “Nath, P.; Yadav, A. K. Anticestodal Properties of Hibiscus rosa-sinensis L. (Malvaceae): An in Vitro and in Vivo Study against Hymenolepis Diminuta (Rudolphi, 1819), a Zoonotic Tapeworm. J. Parasit. Dis. 2016, 40 (4), 1261–1265. https://doi.org/10.1007/s12639-015-0664-2.     

The authors demonstrated that Hibiscus rosa-sinensis leaves in vitro and in vivo possess anticestodal properties against Hymenolepis Diminuta (Rudolphi, 1819), a zoonotic tapeworm. Certainly, the extracts found in plants, whether they are alcoholic, aqueous, or ethereal extracts, are the reflection of all the plant dust, the extracts only dissolve and concentrate the secondary metabolites. The writing was improved.

Reviewer: Ln 165-170: The information in here may cause a concern of using Hibiscus forage meal as a feed ingredient due to the potential negative impact on reproduction performance. For example, the reduction of estrogenic activity during pregnancy and abortion are not considered as a good response to female animals. Comprehensive discussion is required.

Authors: The wording has been improved. However, future work with reproductive females is necessary to specify at what levels of forage consumption of Hibiscus rosa-sinensis, and at what moments of the reproductive life, the aforementioned hormonal changes and reproductive damage mentioned by 5, 57 and 58.

Reviewer: Ln 174-177: Please revise the sentence. “Working as an anti-inflammatory” is not a clear message, probably a noun is missing in this part of the sentence.

Authors: Done. The wording of the sentence was improved

Reviewer: Ln 183-185: Please revise the sentences

Authors: Done. Sentence wording improved.

Reviewer: Ln 208: Please revise the sentence. What types of digestibility is improved by including Hibiscus meal?

Authors: Done. Sentence wording improved.

Reviewer: Ln 212-215: Message is not clear. Two sentences can be written into one sentence and clearly state the comparison.

Ln 217-218: star grass+Hibiscus system showing a better growth rate than star grass+Morus alba and star grass+ Gliricidia sepium systems. Please report the proportion of star grass and Hibiscus

Authors: Done. Sentence wording improved.

Reviewer: Ln 226: Viet Nam should be one Word

Authors: Done.

Reviewer: Ln 228: Please check the intake in the reference, fresh foliage intake was not reported as 2.49 kg/d in the paper published by Nhan (1998).

Authors:  We read the article again, and the information provided in the manuscript is correct (2.49 kg/d), it seems you are referring to a similar article by the same author.

Reviewer: Ln 229: When feed intake and gain of goats fed Hibiscus rosa-sinensis were lowest compared with goats fed Sesbania grandiflora, Leucaena leucocephala, and Ceiba pentadra, how can goat producers be confident of feeding Hibiscus rosa-sinensis as a feed ingredient to goat?

Authors: This is justified, because although the 90-day regrowth of Hibiscus forage, with a low crude protein content (13.89%), the goats gained 77 g/animal/day. If a forage less age of 30 to 60 days of regrowth was used and therefore from 16.6 to 19.6% of CP, surely the response would have been much higher (see table 3 of the review)

Reviewer: Ln 234-235: More information on what area should is required? Please discuss it.

Authors: The discussion of the results is increased. It is important noted, as we have mentioned, there are few scientific articles that address the use of this plant. We are convinced that this manuscript will be the starting point for further studies to make this plant an effective alternative in animal nutrition.

Reviewer: Ln 236: The information presented in rabbits feeding does not support that feeding Hibiscus meal to growing rabbit can go up to 25% in a diet in table 6. Please include the information for growing rabbit.

Authors: The wording was improved.

Reviewer: Ln 247-248: This sentence is not a complete sentence. Please revise it.

Authors: Done

Reviewer: Ln 249-257: This is a very long sentence and hard to read. Please revise it.

Ln 261-266: Another long sentence. Please revise it.

Authors: Done

Reviewer: Ln 271-274: The positive responses on growth of chicken, carcass, economic benefits are in comparison with what dietary treatments?

Authors: The basal diet was a traditional balanced feed for organic chickens, without hibiscus flour, with excellent nutritional value.

Reviewer: Ln 276: Based on one chicken study, the results should not be used to recommend that Hibiscus meal in other poultry diets.

Authors: Done

Reviewer: Ln 277-283: Another long sentence. Please revise it. No difference on nutrient and energy digestibility of 0 and 10% of Hibiscus rosa-sinensis forage meal should not be viewed as no difference on growth performance of pigs

Authors: Done. We change the wording

Reviewer: Ln 285: Please revise “Safe inclusion levels” as a section title because safety assessment is much more complicated than just conducting a growth performance feeding trial. Safety assessment requires more comprehensive evaluation.

Authors: Done. We change the wording

Reviewer: Ln 289: feeding recommendation for poultry in Table 6 was based on one organic chicken feeding trial, so the authors should not conclude that 5% and 10% is the safe levels to use in poultry starter and grower. Similarly, the apparent ileal digestibility study conducted by Ly (2005) should not be used as forms and safe levels of use of Hibiscus forage in swine feed. The assays evaluating safety of feed ingredients fed to pigs is beyond nutrient digestibility. Until more studies are reported and validated, recommendation for safe inclusion should not be concluded.

Authors: Done. We change the wording

Reviewer: Ln 301: “Information con the use of Hibiscus rosa-sinensis as feed,” is “con” a typo?

Authors: Done. We change the wording

Reviewer: Ln 301-308: This is a redundant information in the conclusion because it has stated in previous section and in table 6. Also, the recommendation of inclusion rate of Hibiscus rosa-sinensis in many species is solely based on limited studies, which should not be viewed as a feeding guideline.

Authors: Done. We change the wording

Reviewer: Ln 317: Authors should double check all the references. They are not consistent; for example: some titles are capitalized, and some are in lowercase. Some references seem like non-peer-reviewed papers and citing them as references in a review paper could be an information concern for publishing on a potential feed ingredient for animals on an international, peer-reviewed, open access journal.

Authors: Done. References were corrected.

Round 2

Reviewer 1 Report

The authors did an adequate job in revising their manuscript. The revised manuscript is acceptable.

Reviewer 3 Report

Ln 139: Some units of chemical composition in table 4 are presented as g/100g and I suggest changing to % for being consistent with others.

Ln 142: OM: materiel is not spelled right. I also think that ND refers to "not detected" is not an accurate term. It should be not determined because the energy levels should be detectable. 

Ln 334: I understand the interest in producing organic products by using natural ingredients. What is the yield of Hibiscus under organic farming? I assume the organic Hibiscus will be consumed by humans instead of livestock animals.

Ln 343: un? Please double check the sentence. 

Ln 355-357: I suggest removing "10% inclusion of this forage is not toxic to pigs," and keeping the main context in voluntary feed intake.  

Ln 364: Please check the spelling for a few words in table 6. (convencional and acording)

Ln 371:  Please double check the spelling (antimplantation)

Ln 388: Please double check the spelling (eficiency)

Ln 388: "Rabbits use this forage (Hibiscus) with eficiency when is supplement ad libitum." The meaning is not clear. Please revise the sentence.

Ln 390: The "efficient" may not fit the message you want to deliver.  

Author Response

Dear reviewer,

Thank you very much for your detailed comments on our manuscripts.  The changes were marked in yellow.

Reviewer: Ln 139: Some units of chemical composition in table 4 are presented as g/100g and I suggest changing to % for being consistent with others.

Authors: Done.

Reviewer: Ln 142: OM: materiel is not spelled right. I also think that ND refers to "not detected" is not an accurate term. It should be not determined because the energy levels should be detectable. 

Authors: Done. The abbreviation was changed.

Reviewer: Ln 334: I understand the interest in producing organic products by using natural ingredients. What is the yield of Hibiscus under organic farming? I assume the organic Hibiscus will be consumed by humans instead of livestock animals.

Authors: The idea was further argued, and a new USDA reference was included. As far as we know, the use of hibiscus in humans is purely medicinal.

 Reviewer: Ln 343: un? Please double check the sentence. 

Authors: Done.

Reviewer: Ln 355-357: I suggest removing "10% inclusion of this forage is not toxic to pigs," and keeping the main context in voluntary feed intake. 

Authors: Done.  

Reviewer: Ln 364: Please check the spelling for a few words in table 6. (convencional and acording)

Authors: Done.

Reviewer: Ln 371:  Please double check the spelling (antimplantation)

Authors: Done.

Reviewer: Ln 388: Please double check the spelling (eficiency)

Authors: Done.

Reviewer: Ln 388: "Rabbits use this forage (Hibiscus) with efficiency when is supplement ad libitum." The meaning is not clear. Please revise the sentence.

Authors: Done

Reviewer: Ln 390: The "efficient" may not fit the message you want to deliver.  

Authors: Done

Round 3

Reviewer 3 Report

The authors have done a good job revising the manuscript compared with the first version. I do not have any further comments to make. I accept in the present form.